# Risk Factors for Cancer Mortality in Spain: Population-Based Cohort Study

**DOI:** 10.3390/ijerph19169852

**Published:** 2022-08-10

**Authors:** Antonio D. Lázaro-Sánchez, Asunción Juárez Marroquí, Jose Antonio Quesada Rico, Domingo Orozco-Beltrán

**Affiliations:** 1Medical Oncology Service, University Hospital of Sant Joan d’Alacant, 03550 Sant Joan d’Alacant, Spain; 2Department of Clinical Medicine, University Miguel Hernández de Elche, Ctra. Nnal. 332, s/n, 03202 Elche, Spain

**Keywords:** cancer, risk factors, mortality, epidemiology

## Abstract

Background: Cancer is considered a major public health problem due to its increasing incidence and high mortality. This study aims to identify risk factors for cancer mortality in Spain. Methods: Retrospective population-based cohort study in 20,397 participants of the 2011/2012 National Health Survey in Spain. Risk factors associated with mortality due to neoplasm from 2011 to 2017 were analyzed, and hazard ratios were calculated with a multivariate Cox model with competing risks for mortality from other causes. Results: Myocardial infarction, chronic obstructive pulmonary disease, cirrhosis, and mental disorders were associated with an increased risk of mortality due to neoplasm. Male sex, age over 50 years, history or current smoking habit, negligible intake of legumes, and poorer self-perceived health were also associated with increased cancer mortality. Conclusions: Comorbidities, tobacco use, poor diet, and worse self-perceived health were the main risk factors for cancer mortality in Spain.

## 1. Introduction

The International Agency for Research on Cancer (IARC) estimated that in 2020, there were around 19.3 million new cases of cancer diagnosed globally and 9.9 million deaths [1,2]. Of these, the European Union accounted for 2.7 million cases and 1.3 million deaths [3]. Cancer mortality worldwide is increasing, while in Europe it has decreased in recent decades, although not for all tumors [4]. In Spain, cancer is one of the main causes of morbidity and mortality, with an estimated 276,239 new cases of cancer for the year 2021 [5].

According to the World Health Organization (WHO), a third of cancer deaths can be collectively attributed to tobacco, alcohol, infections, sedentarism, and inadequate diets [6]. Tobacco is the most consistent risk factor for lung cancer, increasing an individual’s risk by 10 to 20 times depending on exposure [7]. It is also implicated in the development of leukemia as well as tumors of the head and neck, esophagus, pancreas, liver, stomach, cervix, kidney, large intestine, bladder, breast, and prostate [6]. Alcohol is responsible for more than 3 million cancer deaths annually worldwide and plays a causal role in the development of oral carcinomas and cancers of the larynx, oropharynx, esophagus, liver, colon and rectum, and breast [8]. Furthermore, there is no safe level of consumption, and a synergistic effect exists with smoking [8,9]. Regarding infections, the IARC classifies 10 microorganisms as carcinogenic [10], accounting for 25% of tumors in low- and middle-income countries [2]. Helicobacter pylori, human papillomavirus, hepatitis B virus and hepatitis C virus are the main causes of cancer attributable to infections [10]. For its part, sedentarism is associated with 5% of cancer-related deaths [11,12], while obesity causes 20% of neoplasms [13]. In contrast, physical activity is associated with a decreased risk of colon, breast, prostate, and endometrial cancer [14]. Finally, the consumption of red meat increases the risk of colorectal cancer, whereas a diet rich in fruits and vegetables shows a modest protective association against this type of tumor [15].

Deepening our knowledge of the risk factors associated with the development of tumors is essential when it comes to preventing and improving the prognosis of cancer. However, the few population-based studies available in the literature tend to focus on selected risk factors for specific neoplasms [7,8,9,10,11,12,13,14]. In addition, most have been carried out in countries other than Spain.

The objective of this study is to identify and quantify risk factors for cancer mortality in Spain.

## 2. Methods

### 2.1. Study Population

This population-based, retrospective, observational cohort study drew data from the 2011/2012 National Health Survey (ENSE11/12), performed by the National Institute of Statistics (INE) [16] from July 2011 to June 2012, using a stratified three-stage complex sampling design with 21,007 participants, representative of all residents in Spain over 15 years of age.

### 2.2. Baseline Measurements 

All respondents from the ENSE11/12 were included in the study. The ENSE11/12 is a large health survey representative of the whole of Spain and is composed of 148 questions in which various sociodemographic and behavioral variables were collected together with data on comorbidities and use of health services. The 2011/12 survey was chosen to obtain sufficient follow-up, and the participants were followed for six years to monitor mortality. To do this, the INE carried out the probabilistic cross-linkage of the ENSE11/12 data with the national vital statistics registry from 2011 to 2017. The outcome variable in this study is death due to neoplasm (code C00-D49 of the 10th edition of the International Classification of Diseases (ICD-10)) and death from any other cause occurring during the follow-up period. Missing values have been eliminated in the analysis variables, and when the number of missing values was greater than 1% of the total, a Not Available (NA) category was created in some explanatory variables.

### 2.3. Statistical Analysis

A descriptive analysis was performed by calculating the frequencies of categorical variables. The factors associated with mortality due to neoplasm were analyzed using contingency tables and the chi-squared test.

To estimate the magnitude of the 6-year mortality risk due to neoplasm, a survival analysis was performed by fitting a multivariate Cox model. This model presents 4 additional characteristics to a standard Cox model: (1) Competitive risks approach between mortality from neoplasms and from other causes, using the approach of Putter et al. [17], as applied by Moore [18]. This characteristic must be taken into account in the analysis, since both types of mortality are competitive causes among themselves. (2) Adjustment of time-dependent variables that allows modeling those variables that violate the proportional hazards hypothesis of the standard Cox model. (3) Weighted analysis to obtain representative estimates of the Spanish population. A complex sampling was applied using as a weighting factor the survey elevation factor divided by its mean, thus obtaining weights centered on the mean [19]. (4) Internal validation of the model to obtain indicators of the predictive capacity of the model, such as the C-index and its 95% CI. For this purpose, the model was fitted to a random test sample composed of 70% of the original sample, and validation was performed with a random test sample of 30% of the original size. The analyses were carried out using the R v.4.0.2 program (R Core Team, Vienna, Austria).

## 3. Results

Of the 21,007 respondents, 20,397 were analyzed, eliminating 610 (2.9%) because of missing values in the analysis variables (less than 1% of the total sample). A category with Not Available (NA) values has been added in variables with more than 1% NA: Social Class (NA = 538, 2.6%), Body Mass Index (NA = 1431, 6.8%), Net Monthly Income (NA = 5441, 25.9%) and Mental Health (NA = 240, 1.1%). Most (58.9%) participants were less than 50 years old, and 5.5% were 80 years or older. Just under half (48.7%) were men, 15.4% were obese, 19.5% were ex-smokers, 27.1% were current smokers, 44.3% were low-risk drinkers (intake of ≤20 g/week in women or ≤40 g/week in men) and 2.5% risk drinkers (>20 g/week in women and >40 g/week in men). High blood pressure was present in 21% of the sample and diabetes mellitus in 7.2%. The descriptions of the rest of the variables in the sample, as well as the six-year cumulative incidence rates for mortality due to neoplasm and mortality from other causes, are shown in Table 1 and in the electronic Appendix A.

The mean follow-up time was 75.1 months (6.2 years), and a median of 77 months (6.4 years). In total, of the 20,397 included participants, 340 (1.7%) died from cancer, 767 (3.8%) died from another cause, and 19,290 (94.6%) survived to the end of the six-year follow-up. Table 2 presents the risks estimated using the multivariate competing risks Cox model for cancer mortality and mortality from other causes. Variables associated with a significantly higher risk of cancer mortality were myocardial infarction (HR 1.37, 95% CI 1.04–1.82), chronic obstructive pulmonary disease (COPD) (HR 1.61, 95% CI 1.29–2.01), cirrhosis or liver dysfunction (HR 1.62, 95% CI 1.10–2.37), mental disorders other than anxiety or depression (HR 2.01, 95% CI 1.47–2.75), male sex (HR 1.56, 95% CI 1.29–1.88), age older than 50 years (HR 8.99 to 45.58, depending on the age group), current smoking habit (HR 1.83, 95% CI 1.16–2.89), past smoking habit (HR 1.44, 95% CI 1.04–2.01), negligible intake of legumes (HR 2.28, 95% CI 1.03–5.02), and worse self-perceived health (HR 2.50 to 3.91, depending on the category). Variables showing a protective association against cancer mortality were higher levels of physical activity during leisure time (HR 0.50 to 0.57); osteoarthritis, arthritis or rheumatism (HR 0.52, 95% CI 0.39–0.70); gastrointestinal ulcer (HR 0.74, 95% CI 0.56–0.96), and chronic allergy (HR 0.66, 95% CI 0.50–0.85).

By region, people in Andalusia (HR 2.19, 95% CI 1.40–3.42), Castilla y León (HR 2.09, 95% CI 1.31–3.33), Asturias (HR 2.09, 95% CI 1.28–3.41), and La Rioja (HR 2.07, 95% CI 1.24–3.45) presented an elevated risk of cancer mortality, while the Basque Country (HR 1.70, 95% CI 1.04–2.78), the Valencian Community (HR 1.63, 95% CI 1.03–2.59), and Murcia (HR 1.00) showed a lower risk.

The multivariate model showed two time-dependent risk factors associated with cancer mortality: an MRI in the previous 12 months and diagnosis of a malignant tumor (Table 2). In the case of the MRI, the log-risk of mortality from neoplasms was estimated at 0.771 − 0.013 × T, where 0.771 is the coefficient of malignant tumor and −0.013 is the coefficient of the interaction with the follow-up time T. Therefore, the risk of death at the beginning of follow-up is estimated at HR = 2.1 (exp [0.771 − 0.013 × 1]) for those who underwent an MRI compared to those who did not, and this risk decreases throughout follow-up. Similarly, for diagnosis with a malignant tumor, the log-risk is estimated at 1.247 − 0.017 × T, based on an HR of 3.4, indicating greater risk of death in those diagnosed versus not diagnosed with a malignancy, and decreasing the risk over time (Figure 1).

The competing risk model indicates significant effects on mortality from other causes, due to age (age: others causes); tobacco use (tobacco: others causes); and having osteoarthritis, arthritis or rheumatism (osteoarthritis: others causes), as presented in Table 2. These HRs express the difference in the effect of these variables for cancer mortality and mortality from other causes. Thus, for the 50–59 and 60–69 year age groups, the risk of death from other causes was 75% and 66% lower than the risk of death from neoplasms, respectively. In other words, people in these age groups had a higher risk of death from neoplasms than from other causes, while having osteoarthritis, arthritis or rheumatism brought a greater risk of death from other causes than from neoplasms.

The model was estimated in a test sample of 14,191 people, with 302 deaths from neoplasms and 738 from other causes, presenting a good fit to the data (LRT χ^2^ = 2471, *p* < 0.001). The model was validated in the test sample of 6206 participants, with 128 deaths from neoplasms and 311 from other causes, obtaining a good predictive capacity with an honest C-index of 0.90 (95% CI 0.88–0.93).

## 4. Discussion

### 4.1. Main Findings

The participants in our study at the highest risk of death due to cancer were those who had myocardial infarction, COPD, cirrhosis or liver dysfunction, mental disorders, an age above 50 years, male sex, ex-smokers or current smokers, negligible intake of legumes, and a worse state of self-perceived health. On the other hand, physical activity during leisure time was associated with less mortality, as was having osteoarthritis, arthritis or rheumatism; gastrointestinal ulcer, or a chronic allergy.

### 4.2. Comparison with Other Studies

A large, recent study in the USA showed that men had a lower incidence rate ratio (IRR) of cancer (IRR 0.96), but worse survival (HR 1.56) compared to women [20]. This finding coincides with the results from our cohort, where we observed a 56% higher risk of cancer mortality in men compared to women. The literature suggests that genetic and environmental factors (e.g., outdoor occupations more frequent in men and therefore greater exposure to solar radiation) play an important role in the differences by sex [21,22,23].

Age is frequently associated with an increased risk of cancer [24]. Several publications confirm an increased risk of breast, prostate, lung, and colorectal cancer after the age of 70 [25]. We observed a 46-fold increased risk of mortality due to neoplasm in patients aged 80 years or older. The accumulation of senescent cells and an aging immune system are some of the factors that favor a pro-tumor environment in the older population [26].

Tobacco is the first isolated cause of preventable morbidity and mortality, with a higher overall relative risk (RR) of cancer in smokers than in non-smokers (RR 1.86) [27,28]. According to our data, past or current smoking was associated with 44% and 83% higher risk of cancer death, respectively, compared to never smoking.

A 2020 systematic review found a strong association between higher levels of physical activity and a reduction of 10% to 20% in the RR of bladder, breast, colon, endometrial, renal, and gastric cancer [14]. In our retrospective cohort, physical activity was associated with a 42% to 50% reduction in the risk of mortality due to neoplasm, depending on frequency of physical activity, compared to sedentary behavior. Several mechanisms have been proposed to explain the possible protective effect of physical activity, including reduced levels of circulating insulin and hormones and improved immune function [29].

The results of this study show that participants who rarely or never consumed legumes had about twice the risk of mortality due to neoplasm than those who consumed these products daily. These data are in line with those published by the PREDIMED study in 2018, which argues that the protective effect of legumes is due to the high content of polyphenols (HR 0.51) [30]. Another recent meta-analysis also associated a diet rich in isoflavones with a reduced risk of endometrial cancer (odds ratio 0.81) [31].

In this study, people with myocardial infarction had a 37% higher risk of mortality due to neoplasm compared to those without. Other work suggests that these people have a modest 5% to 8% increased risk of cancer [32,33]. Recently, a large Norwegian prospective cohort demonstrated a higher risk of cancer in the first 6 months (HR 2.15) and at three years (HR 1.60) after a myocardial infarction. Colorectal, prostate, and lung cancer were the most frequently diagnosed neoplasms [34]. Current evidence suggests that MI and cancer share the same molecular pathways of disease development and progression [35].

People in our cohort who reported having osteoarthritis, arthritis or rheumatism presented half the risk of cancer death compared to those without these comorbidities. The ENSE 11/12 did not distinguish between osteoarthritis and arthritis, which limits the interpretation of the results. However, the literature suggests a 20% lower risk of breast and colorectal cancer in patients with a history of rheumatoid arthritis, without finding a solid cause to justify it [36,37]. A Spanish study from 2008, on the other hand, estimated a higher standardized incidence ratio (SIR) of cancer (SIR 1.23) in patients with rheumatoid arthritis, at the expense of an increased risk of leukemia (SIR 8.8), lymphoma non-Hodgkin (SIR 5.4) and lung adenocarcinoma (SIR 3.5). Regarding osteoarthritis, no data are available in the literature that relate this pathology to the risk of mortality due to neoplasm [38].

In our cohort, people with chronic allergy were shown to have about a 34% lower risk of mortality due to neoplasm than those without. The potential protective effect of allergy on certain neoplasms (pancreatic cancer RR 0.82, glioma RR 0.61, colon HR 0.76, rectum HR 0.54) has been previously observed, suggesting a possible antitumor effect of IgE-mediated immunity [39,40]. However, the evidence is limited, and studies that take into account different biological markers are required.

In this study, COPD was associated with a 61% increased risk of mortality due to neoplasm. It is generally accepted that chronic inflammation plays a key role in the pathogenesis of lung cancer in COPD patients, increasing its incidence two- to five-fold [41,42].

Participants with a history of gastrointestinal ulcer had a 26% lower risk of mortality due to neoplasm compared to those without this condition. Other authors have reported similar results, speculating that duodenal ulcer may have a protective effect against gastric cancer (0.6%) through the preservation of acid secretion values [43]. However, a 2014 study in the USA suggests that this “protection” is rather an artifact that reflects the differences in the extent of *Helicobacter pylori* gastritis. On the other hand, studies consistently describe gastric ulcer as associated with an increased risk of cancer in the same location (SIR 1.8) [44]. In this study, the survey did not distinguish between the two pathologies, which precludes any solid conclusions.

Most patients with hepatocarcinoma have underlying liver cirrhosis [45], although the causal pathway is not clearly defined [46]. A study conducted in Sweden and Iceland in 2011 confirmed that the overall risk for cancer other than hepatocarcinoma was 2–3 times higher in patients with cirrhosis compared to the general population [47]. In this study, cirrhosis or liver dysfunction was associated with a 62% increased risk of mortality due to neoplasm (HR 1.62, 95% CI 1.11–2.37).

Reported cancer incidence in patients with severe mental illness is variable. A US retrospective cohort study from 2012 suggested that adults with severe mental disorders, such as schizophrenia or bipolar disorder, have a higher overall risk of cancer (SIR 2.6), especially lung (SIR 4.1–4.7), colorectal (SIR 3.5–4) and breast cancers (SIR 1.9–2.9). The authors hypothesized that these findings were probably related to behavioral, social, or pharmacological factors that are not fully understood [48]. In fact, other studies confirm disparities in cancer screening and prevention, as well as a suboptimal administration of antineoplastic treatments in this patient group [49]. The participants with mental problems in this study showed twice the risk of mortality due to neoplasm than those without these disorders.

### 4.3. Strengths and Limitations

The findings of this study should be interpreted in light of certain limitations. Data were retrospective, and some survey items on health determinants encompassed several pathologies, which made the interpretation of the results difficult. Moreover, the outcome variables in this study are total cancer deaths and deaths from other causes. Therefore, the risk factors associated with the different tumor types could not be evaluated. Another limitation may reside in predictive and/or confounding factors not measured in the survey. Another limitation could be due to the self-referenced nature of the survey variables, although these data come from official sources, and this is a common problem in all health surveys.

Strengths of the study include its large sample size, which allowed the adequate estimation and validation of the model and the use of the elevation factor from the complex sampling design in the analysis, enabling representative estimates of the Spanish population in 2011. The use of a competing risks approach allows differential identification of the factors associated with mortality due to neoplasm and from other causes.

## 5. Conclusions

There is a greater risk of mortality due to neoplasms in people with myocardial infarction, COPD, cirrhosis or liver dysfunction, mental disorders other than anxiety or depression, age over 50 years, male sex, past or present tobacco use, low intake of legumes, and worse self-perceived health status. In contrast, physical activity; osteoarthritis, arthritis or rheumatism; gastrointestinal ulcers; and chronic allergy are associated with lower mortality from neoplasms.

## Figures and Tables

**Figure 1 ijerph-19-09852-f001:**
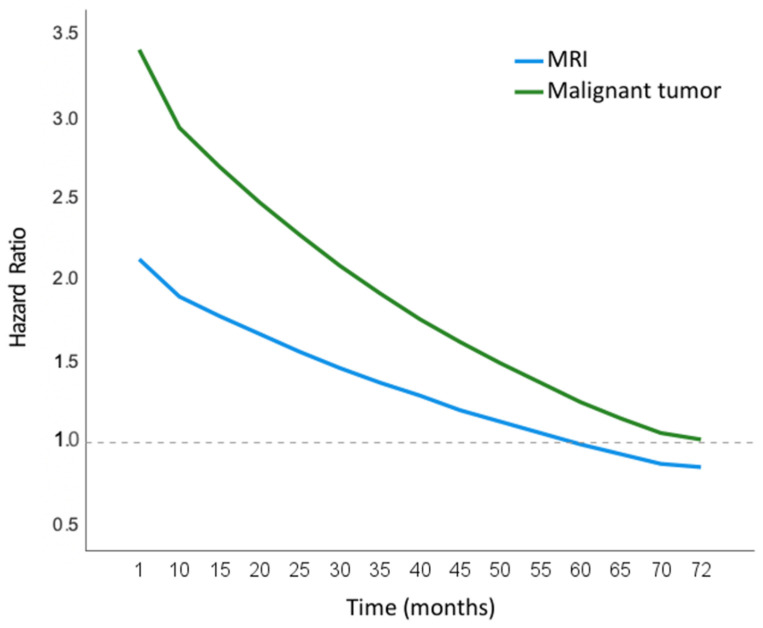
Risk of death (HR) throughout the follow-up period of the two time-dependent variables in the Cox multivariate model: Magnetic Resonance Imaging (MRI) (exp (0.771 − 0.013 × T)) and Malignant tumor (exp (1.247 − 0.017 × T)).

**Table 1 ijerph-19-09852-t001:** Characteristics of the sample (*n* = 20,397).

Variables		*n*	%
Mortality	Survived	19,290	94.6%
	Death due to neoplasm	340	1.7%
	Death from other cause	767	3.8%
Age group (years)	<50	12,015	58.9%
	50–59	3106	15.2%
	60–69	2393	11.7%
	70–79	1767	8.7%
	≥80	1115	5.5%
Sex	Male	9942	48.7%
	Female	10,455	51.3%
Body mass index	Normal (<25 kg/m^2^)	9019	44.2%
	Overweight (25.0 to <30 kg/m^2^)	6803	33.4%
	Obese (≥30 kg/m^2^)	3144	15.4%
	NA	1431	7.0%
Tobacco use	Never smoker	10,908	53.5%
	Ex-smoker	3971	19.5%
	Smoker	5518	27.1%
Alcohol intake	None	10,857	53.2%
	Low risk (≤20 g/week women and ≤40 g/week men)	9029	44.3%
	Risky drinking (>20 g/week women and >40 g/week men)	511	2.5%
Physical activity	Sedentary	9051	44.4%
	Occasional physical activity	6735	33.0%
	Frequent physical activity	2668	13.1%
	Athletic training	1943	9.5%
Vegetable intake	Daily	9566	46.9%
	≥3 servings a week	6990	34.3%
	1–2 servings a week	2761	13.5%
	<1 serving a week	690	3.4%
	Rarely or never	390	1.9%
Intake of legumes	Daily	312	1.5%
	≥3 servings a week	4738	23.2%
	1–2 servings a week	12,161	59.6%
	<1 serving a week	2400	11.8%
	Rarely or never	786	3.9%
Self-perceived health	Very good	4396	21.6%
	Good	10,413	51.0%
	Okay	4065	19.9%
	Bad	1224	6.0%
	Very bad	299	1.5%
Comorbidities	Hypertension	4278	21.0%
	Diabetes mellitus	1465	7.2%
	Malignant tumor	575	2.8%
Health services use	Hospital admission in previous 12 months	1695	8.3%
	Primary care visit in previous 30 days	5719	28.0%
	CT scan in previous 12 months	1536	7.5%
	Ultrasound in previous 12 months	2925	14.3%
	MRI in previous 12 months	1513	7.4%

CT: computed tomography; MRI: magnetic resonance imaging; NA: not available.

**Table 2 ijerph-19-09852-t002:** Multivariable Cox model for mortality due to neoplasm, with competing risks for other causes of death (*n* = 20,397).

Variable		Beta	Error	HR	95% CI	*p* Value
Autonomous communities	Murcia	0	–	1		
	Andalusia	0.785	0.227	2.19	(1.40–3.42)	0.001
	Castilla y León	0.736	0.239	2.09	(1.31–3.33)	0.002
	Asturias	0.735	0.251	2.09	(1.28–3.41)	0.003
	Rioja	0.727	0.261	2.07	(1.24–3.45)	0.005
	Extremadura	0.717	0.252	2.05	(1.25–3.36)	0.004
	Balearic Islands	0.547	0.284	1.73	(0.99–3.02)	0.054
	Basque Country	0.530	0.252	1.70	(1.04–2.78)	0.036
	Canary Islands	0.512	0.264	1.67	(1.00–2.80)	0.052
	Valencian Community	0.490	0.236	1.63	(1.03–2.59)	0.038
	Madrid	0.455	0.240	1.58	(0.98–2.52)	0.058
	Aragon	0.439	0.247	1.55	(0.96–2.52)	0.075
	Catalonia	0.283	0.233	1.33	(0.84–2.10)	0.22
	Castilla-La Mancha	0.273	0.249	1.31	(0.81–2.14)	0.27
	Navarra	0.245	0.286	1.28	(0.73–2.24)	0.39
	Cantabria	0.156	0.267	1.17	(0.69–1.97)	0.56
	Galicia	0.042	0.244	1.04	(0.65–1.68)	0.86
	Ceuta-Melilla	−0.020	0.361	0.98	(0.48–1.99)	0.96
Age group (years)	<50	0	–	1		
	50–59	2.188	0.389	8.92	(4.16–19.11)	<0.001
	60–69	2.848	0.371	17.25	(8.34–35.70)	<0.001
	70–79	3.336	0.378	28.10	(13.39–58.94)	<0.001
	≥80	3.819	0.391	45.58	(21.18–98.09)	<0.001
Sex	Male	0.443	0.095	1.56	(1.29–1.88)	<0.001
Tobacco use	Never smoker	0	–	–		
	Ex-smoker	0.366	0.169	1.44	(1.04–2.01)	0.030
	Smoker	0.603	0.233	1.83	(1.16–2.89)	0.010
Physical activity	Sedentary	0	–	1		
	Occasional physical activity	−0.549	0.094	0.58	(0.48–0.69)	<0.001
	Frequent physical activity	−0.703	0.296	0.50	(0.28–0.88)	0.017
	Athletic training	−0.593	0.247	0.55	(0.34–0.90)	0.017
Intake of legumes	Daily	0	–	1		
	≥3 servings a week	0.075	0.366	1.08	(0.53–2.21)	0.84
	1–2 servings a week	0.045	0.356	1.05	(0.52–2.10)	0.90
	<1 serving a week	0.222	0.377	1.25	(0.60–2.61)	0.56
	Rarely or never	0.824	0.403	2.28	(1.03–5.02)	0.041
Self-perceived health	Very good	0	–	1		
	Good	0.498	0.216	1.65	(1.08–2.51)	0.021
	Okay	0.916	0.221	2.50	(1.62–3.85)	<0.001
	Bad	1.035	0.239	2.82	(1.76–4.50)	<0.001
	Very bad	1.365	0.270	3.91	(2.31–6.64)	<0.001
Physical limitations in previous 30 days	Severely limited	0	–	1		
	Moderately limited	−0.453	0.126	0.64	(0.50–0.81)	<0.001
	Not limited	−0.745	0.140	0.48	(0.36–0.62)	<0.001
Myocardial infarction		0.318	0.143	1.37	(1.04–1.82)	0.026
Arthrosis or arthritis		−0.651	0.153	0.52	(0.39–0.70)	<0.001
Chronic allergy		−0.424	0.135	0.66	(0.50–0.85)	0.002
COPD		0.476	0.113	1.61	(1.29–2.01)	<0.001
Gastrointestinal ulcer		−0.307	0.137	0.74	(0.56–0.96)	0.025
Cirrhosis		0.482	0.194	1.62	(1.11–2.37)	0.013
Mental health problems		0.698	0.159	2.01	(1.47–2.75)	<0.001
Malignant tumor		1.247	0.258	3.48	(2.10–5.77)	<0.001
MRI in previous 12 months		0.771	0.289	2.16	(1.23–3.81)	0.008
Malignant tumors × T		−0.017	0.006	0.98	(0.97–1.00)	0.004
MRI in previous 12 months × T		−0.013	0.006	0.99	(0.98–1.00)	0.039
Age group:	<50	0		1		
others causes	50–59	−1.408	0.504	0.25	(0.09–0.66)	0.005
	60–69	−1.070	0.459	0.34	(0.14–0.84)	0.020
	70–79	−0.304	0.459	0.74	(0.30–1.81)	0.51
	≥80	0.426	0.464	1.53	(0.62–3.80)	0.36
Tobacco:	Never smoker	0		1		
others causes	Ex-smoker	−0.436	0.192	0.65	(0.44–0.94)	0.023
	Smoker	−0.229	0.290	0.80	(0.45–1.40)	0.43
Arthrosis: others causes	Yes	0.489	0.176	1.63	(1.16–2.30)	0.006

T: time of follow-up; HR: hazard ratio; CI: confidence interval; COPD: chronic obstructive pulmonary disease; MRI: magnetic resonance imaging. Adjustment model: *n* test sample = 14,191; mortality due to neoplasm in test sample *n* = 302; mortality due to other causes test sample *n* = 738; LRT χ^2^ = 2471 (*p* < 0.001); Validation model: test sample *n* = 6206; mortality in test sample *n* = 128; mortality due to other causes in test sample *n* = 311; honest C-index = 0.903, 95% CI 0.876–0.930.

## Data Availability

The INE carried out and provided the probabilistic cross-linkage of the ENSE 11/12 with the national mortality registry in 2017. The data related to the variables of the survey can be consulted at https://www.mscbs.gob.es/ (accessed on 5 May 2022).

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
