# Peer review of "Risk Factors for Cancer Mortality in Spain: Population-Based Cohort Study"

_ijerph, 2022, doi:10.3390/ijerph19169852_

Round 1

Reviewer 1 Report

Now it is fine.

Author Response

Thank you very much for your comments and review. We think it has improved the content and writing of the manuscript.

Reviewer 2 Report

Thanks for reviewing the manuscript and providing satisfactory responses

Result:  in relation to the time of follow-up, please specify the more appropriate summary description (mean or median)?

Author Response

Thank you very much for your comments and review. We think it has improved the content and writing of the manuscript. The description of follow-up time is reflected in the Results section, line 113, with the following sentence, mentioning the mean and median follow-up:

"The mean follow-up time was 75.1 months (6.2 years), and a median of 77 months (6.4 years)".

This sentence was added in the first revision, at the suggestion of the reviewers.

Reviewer 3 Report

The objective of this study is to identify and quantify risk factors for cancer mortality 54 in Spain by using Retrospective population-based cohort study in 20,397 participants of the 2011/2012 12 National Health Survey in Spain. Authors concluded that comorbidities, tobacco use, poor diet, and worse self-perceived health were the 19 main risk factors for cancer mortality in Spain.

The study design and statistical analysis are adequate in relation to the research objectives. The results are clearly presented and discussed. In conclusion, the interpretation of the results is not entirely precise. Namely, for variables for which an HR of less than 1 was obtained, it cannot be claimed a priori that they are protective. They were only less frequently present in those who died. Therefore, the conclusion should be corrected.

Author Response

Thank you for your comment. We have modified the last sentence of conclusions, changing the term "protection" with "association with lower mortality", leaving the sentence as follows:

"Instead, physical activity; osteoarthritis, arthritis or rheumatism; gastrointestinal ulcers; and chronic allergy are associated with lower mortality from neoplasms".

This manuscript is a resubmission of an earlier submission. The following is a list of the peer review reports and author responses from that submission.

Round 1

Reviewer 2 Report

The objectives of the study were to identify the factors associated with mortality from neoplasms in Spain. 

Introduction

Authors should remove subtitles from this section.

Methods

Possibly the section, from my point of view, with more weaknesses of the paper. They studied a sample identified from the Spanish National Health Survey corresponding to the year 2011-12. They use the variables collected in this survey (cross-sectional information): sociodemographic, use of health services, lifestyles, etc. This cohort is measured at six years to identify the cause of death by linking the survey information with the INE information. They use Cox models to determine associations. 

Results

Table 1 presents the description of the subjects (n,%) between different variables. In my opinion, the table should be stratified either by gender or by age group. Table 2 presents the HR of the  major covariable . The information is repeated by age and tobacco use without it being clear to the reader to which the authors refer in each case. I don't understand the meaning of Figure 1. Presenting the HR by period of time. I consider an analysis of survival more appropriate. Why the intake of legumes is introduced as a covariate and not other dietary variables?. 

I consider the article with serious methodological and analysis deficiencies to be published. 

Reviewer 3 Report

11. Reading the manuscript I have realized that apparently no relevant literature is existing on risk factors for cancer mortality for Spain yet. That´s a pity and should be carried out. This seems to be the main rationale for this manuscript. On the other hand the results presented by you for Spain are generally well-known and true for most countries worldwide. This is demonstrated in your Discussion, where you are comparing your results with other international results appropriately.

22.  What I am missing is any information how valid and representative your data are. That´s the main reason why I am recommending a major revision. E.g. I am missing the following information:

a.      Is the National Health Survey based on questionnaires? Is the questionnaire validated / published / developed specifically for this Spanish survey? What about the response rate(s) in this survey?

b.     You are saying “… those with missing values for any of the study variables were excluded”. How many are these? Were they definitely excluded from the entire analyses or only for the specific analysis where the corresponding answer is missed? Looking to Table 1, it seems that only for the item “Body mass index” some missing data are existing (7.0%). It seems unlikely, that for all other questions not any missing information are provided. 

c.      Some examples why I am doubting the meaningfulness of the data: Looking in Table 1 to the items “vegetable intake” and “intake of legumes” it is not comprehensible to me that only 1.5% have a daily “intake of legumes”, but 46.9% have a daily “vegetable intake”. And: I can hardly imagine that the majority of the participants (53.5% and 53.2%, respectively) were never smoker or having not any alcohol intake; I think, this percentages must be lower.

33.  Another main objection is that no information is given about the quality of follow-up. I wonder how many individuals are lost-to-follow-up. I cannot imagine, that all 20,937 participants are followed up till 2017 to know whether they have died or not within this 6 year time period. For instance, an information about the mean observation time would be helpful.

44. In Table 2, lines between “myocardial infarction” and “MRI in previous 12 months x T” are unclear:

a.       Some lines are not indented correctly.

b.      Abbreviation “MRI” should be explained.

c.       What does “… x T” mean?

d.      What does “malignant tumor” mean in this context? Is it meant that these participants are died due to a neoplasm (n=340 as shown in the top of Table 1) or is it meant as comorbidity (a second tumor?) (n=575 further down in Table 1)?  

55. Figure 1 is not comprehensible to me.

66. Reference list: “accesed” should be substituted by “accessed” in some places.

Round 2

Reviewer 2 Report

n my opinion there are significant methodological deficits in the article. I do not recommend its publication. These deficits were pointed out in my first review and the authors do not contribute anything new. 

Reviewer 3 Report

Your responses regarding my main objections have been informative to me, but I am still wishing to have more information about the method in the manuscript (also viewed very critically by another reviewer). So I wonder why you don´t add some of the information gave in your response to the manuscript. You are saying that these details are beyond the scope of this manuscript, but without having this information in the manuscript it seems not unpersuasive to me.

I wrote that Figure 1 is not comprehensible to me. Especially because another reviewer have said the same, I think you should think about how to give a better understanding to the readers. But in your response you only referred to lines 134-142. Generally, a figure should be self-explanatory.

Unfortunately, I was not able to see all your changes made in the revised version. So it happened to me only by chance to see that you have removed Main findings in the Discussion (lines 163-168). Why?

Re my point 2b. you have added: “Missing values have been eliminated in the analysis variables, and when the number of missings was high, a Not Available (NA) category has been created in some explanatory variables“. Please, specify what “high” means.